**Data Availability Statement:** Due to Twitter terms of service limitations we cannot provide the text of the Tweets that we have analyzed. However we have provided Tweet ID numbers in supporting information files.

# COVID-19: Retransmission of official communications in an emerging pandemic

**Jeannette Sutton**[1]*, **Scott L. Renshaw**[2], **Carter T. Butts**[2,3,4,5]

**1** College of Emergency Preparedness, Homeland Security, and Cybersecurity, University at Albany, State University of New York, Albany, New York, United States of America, **2** Department of Sociology, University of California Irvine, Irvine, California, United States of America, **3** Department of Statistics, University of California Irvine, Irvine, California, United States of America, **4** Department of Electrical Engineering and Computer Science, University of California Irvine, Irvine, California, United States of America, **5** Institute for Mathematical Behavioral Sciences, University of California Irvine, Irvine, CA, United States of America

* jsutton@albany.edu

## Abstract

As the most visible face of health expertise to the general public, health agencies have played a central role in alerting the public to the emerging COVID-19 threat, providing guidance for protective action, motivating compliance with health directives, and combating misinformation. Social media platforms such as Twitter have been a critical tool in this process, providing a communication channel that allows both rapid dissemination of messages to the public at large and individual-level engagement. Message dissemination and amplification is a necessary precursor to reaching audiences, both online and off, as well as inspiring action. Therefore, it is valuable for organizational risk communication to identify strategies and practices that may lead to increased message passing among online users. In this research, we examine message features shown in prior disasters to increase or decrease message retransmission under imminent threat conditions to develop models of official risk communicators' messages shared online from February 1, 2020-April 30, 2020. We develop a lexicon of keywords associated with risk communication about the pandemic response, then use automated coding to identify message content and message structural features. We conduct chi-square analyses and negative binomial regression modeling to identify the strategies used by official risk communicators that respectively increase and decrease message retransmission. Findings show systematic changes in message strategies over time and identify key features that affect message passing, both positively and negatively. These results have the potential to aid in message design strategies as the pandemic continues, or in similar future events.

## Introduction

As the most visible face of health expertise to the general public, health agencies have played a central role in alerting the public to the emerging COVID-19 threat, providing guidance for protective action, motivating compliance with health directives, and combating misinformation. Social media platforms such as Twitter have been a critical tool in this process, providing

**Funding:** This work was supported by the National Science Foundation grant number CMMI - 2027399 to JS and CMMI-2027475 CTB. The funders had no role in study design, data collection and analysis, decision to publish, or preparation of the manuscript.

**Competing interests:** The authors have declared that no competing interests exist.

a communication channel that allows both rapid dissemination of messages to the public at large and individual-level engagement [1]. For all their virtues, however, these platforms also pose their own challenges. Prominent among these is the problem of successfully reaching members of the public beyond their relatively small circle of immediate followers, a difficulty compounded by the highly crowded online media environment [1]. To reach a large audience, a message must typically be retransmitted by the sender's followers to others on the platform, a process that depends critically on the salience of the message to the user base and users' own decisions to retransmit it or not [2]. While each retransmission event is inevitably idiosyncratic, past research has shown that message style, structure, and content, as well as the sending context, can affect the overall rate at which messages are passed on, providing "levers" that can be exploited to maximize message amplification [3]. Characterizing the specific factors shaping message retransmission in the unfolding COVID-19 pandemic is thus important for evidence-based messaging strategies to be used in the months ahead.

In this paper, we examine message retransmission on one of the most widely used social messaging systems (Twitter), focusing on original messages posted by public agencies responding to COVID-19. Using a census of messages posted by approximately 700 accounts during the first three months of the pandemic's presence in the United States, we evaluate style, content, and contextual factors enhancing or inhibiting message retransmission. As we show, many factors affecting message retransmission rates are similar to those observed for other hazard events; however, the unique circumstances of the pandemic lead to some distinctive patterns of message passing that may inform communication strategies.

## Background

Public health and emergency management agencies are on the front lines of informing and educating the public about the science of virus transmission and prevention [4]. In response to a threat such as COVID-19, their mission requires them to communicate accurate and credible information to local populations using all means of information delivery. Currently, one of the most effective means of *directly* reaching the public is through the use of digital and social media. Social media enables delivery of timely and actionable risk information to the public while also supporting ongoing dialog, potentially increasing trust and reducing fear and erroneous rumors fueled by misunderstanding [5]. This is increasingly important in the case of the coronavirus pandemic, which has now affected all 50 states across the country and requires nearly hourly updates about changing conditions and policies affecting at risk individuals.

During this uncertain time period, online information seeking and sharing is likely to see a dramatic increase, as seen in research on past events [6,7] requiring effective use of online communication infrastructure to alert and inform a public at risk. Informal online communication channels have the potential to serve as a platform for risk communication amplification and to facilitate engagement through dialog. Communication of timely risk information is absolutely vital for behavioral change to protect public health and safety; however, prior work on messaging in response to emerging health threats such as Zika [8] and Ebola [9] has shown that effective messaging strategies can depend upon details of the threat itself, while work on more conventional infrastructure-disrupting hazards such as fires and floods has likewise suggested that providing critical information regarding "hazard-adjacent" events such as closures, outages, and transportation interruptions can be vital for effective communication with the public [3]

In contrast, COVID-19 has affected every state and U.S. territory, resulting in a nationwide state of emergency where multiple levels of response are actively communicating about the risk. In addition to the pervasive threat posed by infection with the SARS-CoV-2 virus itself,

mitigation measures intended to slow the spread of the infection have been complex and disruptive, requiring both individuals and organizations to radically alter or curtail their normal patterns of activity (often at severe personal or economic cost). In a very real sense, every member of the population is both a responder and a victim of the pandemic, whatever their age, occupation, or place of residence. This is a marked departure from more commonly experienced hazards (including public health hazards) in which there is a clear separation between the impacted population, those responding to it, and the remainder of the country, and in which mitigation measures place few demands on those not directly involved in the incident itself. Adding to the difficulty is the fact that social distancing associated with COVID-19 mitigation attenuates informal social contacts that are normally an important source of information and support during crises. As such, the need for official messaging during the COVID-19 pandemic is arguably far greater than in more commonly encountered hazard settings, and the demands on communicators to inform and motivate are higher. With no immediate end to the event in sight, these needs are likely to continue to grow.

In this study we examine two aspects of official communication on Twitter during the first three months of the response (February–April). First, we investigate the messaging strategies used by public agencies in the evolving COVID-19 response. In particular, we investigate how official tweets differ before and after March 13, the day on which the federal government announced a nationwide emergency. Second, we investigate the predictors of message retransmission. We include *message features*, that is, the design choices and the message content that is shared by our sample of official risk communicators. We also investigate the effect of *message senders*, that is, the type of account (public health, emergency management, or elected official) and the level of government involved (local, state, or federal), as well as the size of the follower networks associated with each account. Finally, we include a measure of salience, pre and post announcement of a nationwide emergency, to investigate the effects of time period on message retransmission. As we show, a wide range of practical and technical content types have been important promoters of information passing during the period, rather than any one content type proving to be of critical interest; salience-enhancing tactics such as the use of interrogative and exclamatory language have not been effective in promoting message retransmission, although inclusion of videos has a fairly large effect. These findings may be useful for informing communication strategy as the pandemic continues to unfold.

## Communication about COVID-19

COVID-19 emerged as an international threat in the early part of 2020, as it spread from China into other regions of the world. By early January, the U.S. Centers for Disease Control began to issue public alerts about a novel coronavirus and by the end of January, the World Health Organization announced that the outbreak was a Public Health Emergency of International Concern. On February 11, the WHO announced COVID-19 as the name of the disease that results from the virus (ultimately dubbed SARS-CoV-2). On March 11, the WHO recognized the spread of COVID-19 as a pandemic and on March 13, the White House proclaimed a National Emergency concerning the COVID-19 outbreak.

Epidemic management traditionally has an "urgent but narrow focus: containing the spread of disease and caring for the sick and dying" [10]. Such a focus suggests communication priorities that are limited to alerting populations to effective protective actions that can be taken individually and collectively while official attention is devoted to surveillance and the development and deployment of treatments. Outbreaks, however, while initially acute, may result in multiple waves of transmission as the virus passes from community to community, requiring a longitudinal communication and a sustained effort to reach persons. Moreover, in comparison

to prior pandemics, the COVID-19 pandemic has occurred during a period of substantially greater epidemiological sophistication and public health preparedness. The disease itself—and its infectious agent—were identified relatively quickly following the initial outbreak in Wuhan, with the viral genome being sequenced by late January of 2020 and models for disease propagation spun up almost immediately. This information, combined with early field reports, made it possible to quickly recommend guidance for effective containment and control (successful in e.g. South Korea), or, barring that, mitigation in the form of social distancing measures intended to slow diffusion and reduce the risk of catastrophic failure of healthcare delivery systems. *These measures have been effective*: COVID-19 has been contained in some locales, and in most others its spread has been sufficiently inhibited to allow hospitals to adapt to the patient load (aka "flattening the curve"). However, a side effect of this strategy has been a *considerable prolongation* of the public response, accompanied by a need for *widespread and ongoing collective action* to maintain social distancing measures. Both factors have resulted in an enhanced requirement for communication with the public on an ongoing basis, not only to pass on warnings and alerts, but also to inform, educate, and continuously motivate individuals in their roles as *de facto* responders in the ongoing disaster.

At this time, it is uncertain how long the COVID-19 pandemic will last, and whether additional infection waves will occur, and whether it will end with natural elimination by herd immunity, suppression via immunization, or conversion to endemicity. With future outbreak events potentially looming on the horizon, and the need to sustain intrusive social distancing measures over a long period, public communication will play an every greater role in the public health response. While all channels are vital, reaching the public directly in the current era increasingly necessitates the use of social media.

## Social media and risk communication

Prior work on social media use and message retransmission in the context of hazard response has revealed a number of factors that are frequently involved in determining messaging effectiveness. Here we briefly review some of the key issues pertinent to the COVID-19 case.

### Longitudinal engagement–where salience seems to matter

Engagement has long been identified as a goal of organizational communication on social media [11] and was originally operationalized as two-way dialogic communication [12]. More recently, social media engagement has been conceptualized as a continuum of communication and includes all communicative acts made by an organization such message content choices to build relationships as well as using hashtags (#), including a hyperlink (URL), and direct replies to exchange messages with other users [5].

Research characterizing online engagement among governmental agencies (local Weather Forecast Offices–WFO- for the National Weather Service) revealed various content-oriented strategies, e.g. to build community and inspire action, as well as message-structure strategies, e.g. replies, hashtags, and URLs, both facilitate interaction and dialogue [5]. However, the use of these strategies by WFOs were found to differ relative to the occurrence of a threat, such as being in a period of an active weather warning. For example during nonthreat periods, engagement content included community building and action-oriented messages encouraging followers to get involved in local education or other efforts. Furthermore, in nonthreat periods, messages were more likely to include URLs and images. In contrast, during threat periods, WFOs tended to share messages about current conditions, "nowcasts," and weather warnings; they also increased their number of direct replies [5].

In the context of the first several months of the COVID-19 event in the U.S. context, the most obvious transition point corresponding to an identified "threat period" was the declaration of a national state of emergency in March; this not only validated the emerging consensus regarding the seriousness of the virus threat in the U.S., but also marked a direct change from earlier Federal communications downplaying the risk of a major outbreak. This hence motivates our first research question:

*RQ: how do risk communication messaging strategies change relative to the declaration of a national state of emergency?*

## Amplification/diffusion/retransmission

Message diffusion, in the form of passing messages from one account to another or retransmission by retweeting, is key to amplifying messages in the social media environment [13]. The more that a message is retweeted, the further the message is spread across online networks, allowing more persons to attend to, and possibly act upon, the information being shared. For risk communicators who send messages during an acute threat period, amplification is a primary objective for reaching those who are vulnerable.

Based upon the findings of seven case studies, Vos et al. [9] proposed the Risk Communication on Social Media (RCSM) model, which identifies key factors that influence message passing under conditions of acute threat. These factors include the design of message itself (including the contents and structural features), the characteristics of the account sending the message (such as organization type), the number of followers associated with the account, and the time during which the message was sent. Results from previous studies [2,14] found that messages containing information about the severity of the threat and actionable information were shared at a higher rate than those that did not contain those contents. Additionally, messages containing media, such as an attached image, were also shared more often [9]. Some features that facilitate engagement, such as the inclusion of a URL or direct replies, were found to decrease message passing [3].

Similar results were identified when investigating longitudinal engagement on social media [15]. Returning to the WFO accounts described above, Sutton et al. [3] modeled message retransmission relative to the timing of threat and nonthreat events. They found that actionable and instructive messages and those that included a visual image were highly shared regardless of the time period; however, daily updates in the form of forecasts or current weather conditions involving little uncertainty, as well as message features that increase interaction, such as direct replies and URLs, decreased message passing. Importantly, in threat periods, they also found that warning messages, in particular those that offered no instruction, decreased message passing.

Longitudinal communication by public safety agencies are likely to follow patterns similar to NWS WFOs, where threats emerge and fade, requiring continuous monitoring of the environment and shifting communication tactics to meet public needs. Engagement strategies designed to further relationship building during nonthreat periods are likely to shift during threat periods in order to privilege communication about the severity of a hazard, its potential impact, and recommendations for protective actions. Such strategies are also likely also to affect message retransmission and the ability to amplify key messages. This motivates our second research question:

*RQ: how do risk communication strategies affect message retransmission relative to the declaration of a national emergency?*

## Methods

### Account selection

We identified 690 accounts representing public health, emergency management, and elected officials (see Table 1). Public health accounts were identified by drawing on publicly available lists [16] and subsequent projects on social media risk messaging [9,17].

The accounts for state governors (plus the District of Columbia and Puerto Rico) (n = 52) and state emergency management organizations (n = 52) local mayors (n = 100) and local emergency management agencies (n = 100) for the 100 largest cities in the U.S. were collected by manually searching organization websites for associated social media accounts.

### Data collection

Between 1 February 2020 and 30 April 2020, we collected 149,335 tweets produced by the Twitter accounts associated with 690 accounts offices using the Twitter Representational State Transfer (REST) API, which allows data to be collected on specific accounts including account

Table 1. Targeted accounts, keywords, and message features; definitions, descriptive information, and examples.

| Variable | Definition | Descriptive information (n, % of total) | Example |
|---|---|---|---|
| **Contents** | | | |
| Susceptibility | | 13,269, 9% | Vulnerable, risk, unlikely, travel, veteran, older, kids, age-60, chronic, immune, dialysis, diabetes, homeless, jail, shelter, facilities, African American |
| Surveillance | Keywords describing strategies to identify population impact | 27,465, 18% | Test, result, case, presumptive, death, contact trace, hospitalize, dashboard, sadden, recover |
| Symptoms | Keywords describing symptoms of disease | 3,855, 3% | Symptom, shortness of breath, fever |
| Efficacy | Keywords instructing individuals on how to protect themselves from the threat | 28,324, 19% | Stay home, self isolate, physical distance, social distance, quarantine, shelter in place, face, mask, hand wash, soap and water, 20 seconds, six feet, disinfect |
| Collective efficacy | Keywords reflecting the capacity to achieve an intended effect | 15,175, 10% | Neighbors, united, solidarity, together, community, mitigate the spread, flatten the curve, stay home save lives, shelter in place |
| Technical information | Keywords describing mechanism of how the virus spreads | 7,973, 5% | Droplet, cough, sneeze, surface, transmission, infect, incubate, contagious, |
| Official Response | Keywords about governmental responses to COVID-19 and how to access information | 28,677, 19% | Public health authority, official, task force, declaration, proclamation, executive order, activate, monitor, model, advisory |
| Information Sharing | Keywords that express | 17,044, 11% | Helpline, hotline, briefing, update, resource guide, webinar, town hall |
| Resilience | Keywords that express thanks and appreciation | 5,517, 4% | Hero, salute, thank, recognize, grateful |
| Closures and openings | Keywords about suspension or reinstatement of service, activities, and facilities | 18,389, 12% | Suspend, close, mandatory, lockdown, visitation, cancel, large gatherings, non essential |
| Primary threat | Keywords used to describe COVID-19 | 17,568, 12% | Coronavirus, COVID-19, ncov, outbreak, pandemic |
| Secondary threat | Keywords used to describe additional threats that result from the pandemic | 21,766, 15% | Mental health, substance abuse, domestic violence, evict, food insecure, blood drive, scam, rumor, stigma, school, unemployment panic buy, PPE, compliance, grief |
| Off topic | Keywords found in off topic messages | 12,403, 8% | Superbowl, state of the union, go red for women, holiday, wx, weather, groundhog |

(*Continued*)

**Table 1.** (Continued)

| Variable | Definition | Descriptive information (n, % of total) | Example |
|---|---|---|---|
| **Structural variables** | | | |
| Photo | Messages coded for the presence of an image or media | 66,830, 45% | World Health Organization @WHO BREAKING "We have therefore made the assessment that #COVID19 can be characterized as a pandemic" - @DrTedros #coronavirus Picture Here 9:26 AM Mar 11, 2020 70.8K Retweets 58.4 Likes |
| Video | Messages coded for the presence of a video clip that has been uploaded or embedded into a Tweet | 6,161, 4% | Ohio Dept of Health @OHdeptofhealth Social distancing works. We are all #InThisTogetherOhio. Embedded Video 22.2M Views 7:00 AM Apr 9, 2020 51.2K Retweets 75.1K Likes |
| Hyperlink / URL | Message contains a hyperlink to external website | 73,091, 49% | All Hands on Deck! Geospatial mapping meets outbreak control. To learn more about the vital role geospatial science and technology can play in public health, go to **https://t.co/lIK3Iarc9o** #CDCEHblog **https://t.co/B2lEA98OHY** |
| Reply | Message is in response to a Tweet from another user | 37,592, 25% | **@Cindy_Lee_G @NCCommerce** Hi, please see this link: https://t.co/vX6KfvO5Og. It includes information as well as contact information for @NCCommerce. |
| Mention | Message includes the Twitter handle of an individual or organization | 49,472, 33% | Thank you **@TaosSkiValley!** #AllTogetherNM https://t.co/t0yFy9n3k8 |

(*Continued*)

**Table 1.** (Continued)

| Variable | Definition | Descriptive information (n, % of total) | Example |
|---|---|---|---|
| Hashtag | Message includes a #keyword hashtag | 67,876, 45% | Our COVID-19 site has information for businesses about how to prepare and what to do if an employee becomes sick. https://t.co/iZI0IsUjWA **#COVID19 #AZTogether** |
| Quote | Message quotes another message in it's entirety | 14,372, 10% |  |
| **Sentence Style** | | | |
| Exclamatory | Message includes an exclamation mark (!) | 22,567, 15% | Stay Healthy Nevada! #StayHomeForNevada #COVID19 https://t.co/8CXK2sJcda |
| Interrogatory | Message includes a question mark (?) | 10,090, 7% | Do you have questions about tenant rights and the current eviction moratorium? Register now for the A Way Home for Tulsa webinar on tenant rights during the COVID-19 pandemic. The webinar will be held Friday, April 3 at 9:30 a.m. #Tulsa https://t.co/IPlYOhcGEk |
| **Organization** | | | |
| Public Health | Public health accounts (international, national, state, and local) | 70,007, 47% | |
| Elected Official | | | |
| Governor | Governor accounts from all 50 states; D.C. and P.R. | 25,139, 17% | |
| Mayor | Mayors of top 100 most populated cities | 28,912, 19% | |
| Emergency Management | | | |
| State | State EM accounts from all 50 states; D.C. and P.R. | 8,422, 6% | |
| Local | Local EM accounts of top 100 most populated cities | 16,855, 11% | |
| **Period Effects** | | | |
| Months | | | |
| February | | 27,342, 18% | |
| March | | 59,090, 40% | |
| April | | 62,903, 42% | |
| National Emergency Declaration | | | |
| Before | | 43,767, 29% | |
| After | | 105,568, 71% | |

and tweet metadata, prior messages posted by the account, and follower count at time of query, and other features (Fig 1). Each targeted account was queried at least once per 24 hours during data collection, allowing retrieval of the complete text of all messages posted during the period, as well as the exact time at which they were posted. On average, each account produced 216.43 messages during the study period, with 12 accounts producing only a single tweet and @WHO producing the most (n = 2624).

### Topic lexicon development

Lexicons—specialized keyword lists that identify topics in the context of a specific corpus— have been developed in previous research by identifying crisis-specific keywords that can be used for monitoring twitter to detect information relevant to a disaster as it is emerging [18,19]. Such systems have generally focused on extracting features from the message, and classifying each tweet based upon user-defined categories. Because we selected targeted accounts belonging to state and local agencies, we were most interested in identifying keywords within our existing corpus. Our goal was not to develop a lexicon that could be utilized on a different set of accounts or applied to a different set of hazards. Instead, we wanted to identify keywords that aligned with 1) theoretical concepts, 2) drew from previous research on Twitter messages in disaster, and 3) observed content and language use during the COVID-19 event.

Keywords (see Table 1) were identified manually by reviewing a random sample of 100 tweets per day over the 90-day data collection period (*n* = 9,000) to identify words or phrases commonly used by our targeted accounts to communicate about the pandemic and associated topics. We began by reviewing past work on message retransmission on Twitter in the context of acute threat and longitudinal risk communication to identify relevant thematic areas [3,9]. This work also includes fear appeals [20,21] with content themes around individual *susceptibility* and *self efficacy*, and the *severity* of the threat. Because of the collective nature of the pandemic, we also include the keywords for content representing *collective efficacy* [21]. Additionally, we identified keywords associated with the secondary threats that emerged over time as the pandemic continued. And finally, because there were a number of tweets within out dataset that were not about the pandemic, we also identified keywords associated with those off-topic tweets.

### Susceptibility

Susceptibility is a state of being likely to be influenced or harmed by a particular thing [9,22]. Keywords describe individuals or groups at risk of COVID-19 infection including older adults, persons with pre-existing health conditions, or institutionalized populations such as persons in nursing homes or those who are incarcerated as well as homeless persons. Additional keywords relevant to susceptibility include expressions of uncertainty such as low risk, high risk, and increased risk.

### Severity

Severity represents the magnitude, or impact, of a threat [9,22]. Here, we divide severity keywords into two categories: *surveillance* strategies to identify the impact on the population such as test result, presumptive positive, contract trace, and person under investigation and *symptoms* of COVID19, such as shortness of breath, cough, and fever.

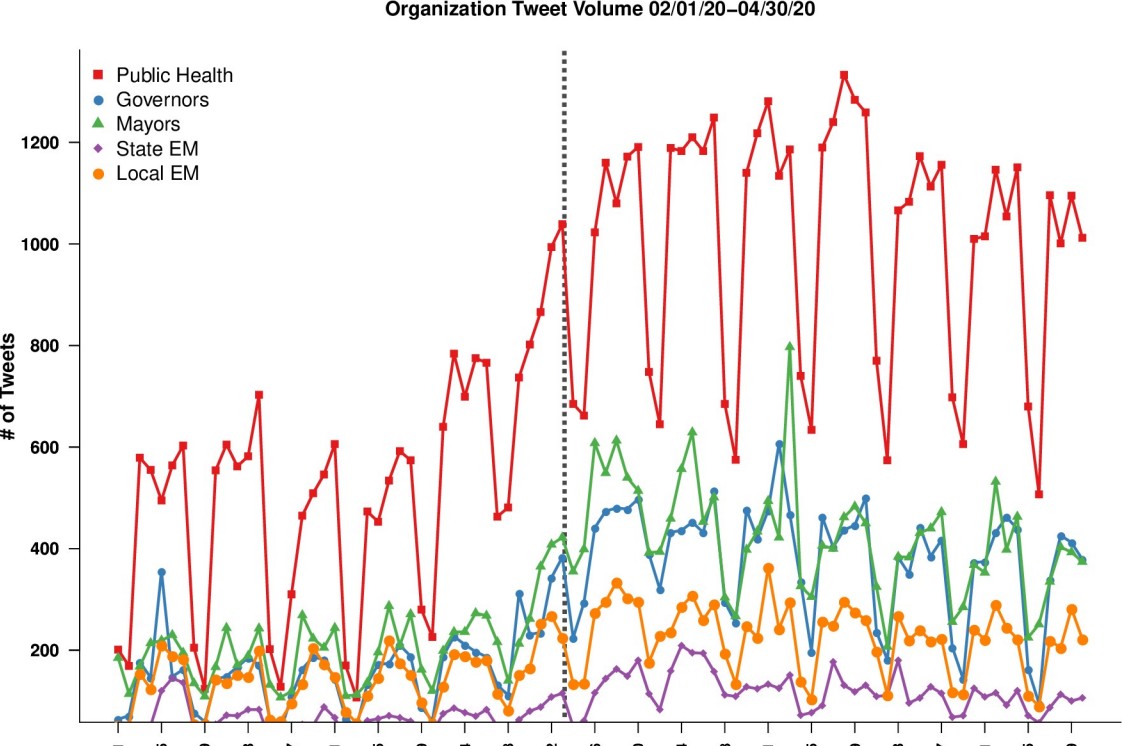

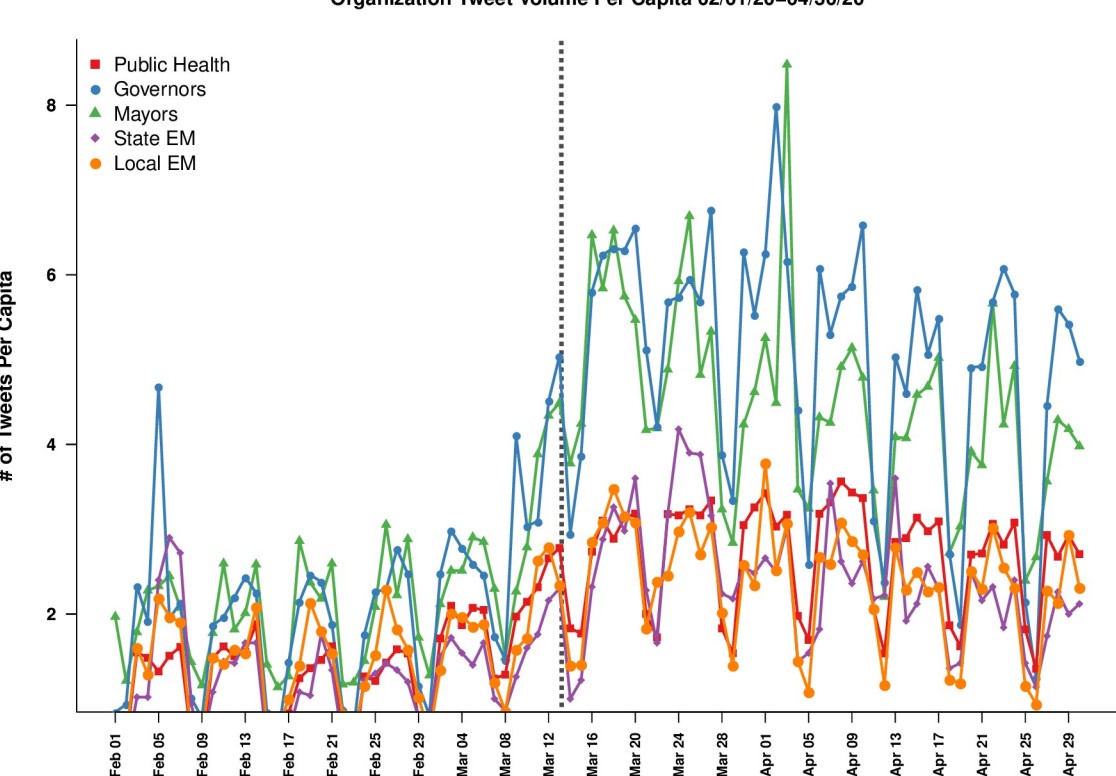

**Fig 1. Daily frequencies of message posting, by account type.** All accounts show a common pattern of weekly variation (with activity peaking mid-week and falling on weekends), with enhanced traffic levels following the Federal emergency declaration on March 13 (dotted line). Total tweet volume (top panel) among the studied organizations is dominated by public health accounts, although per-account activity is higher for elected officials (green and blue lines, bottom panel). These differences are controlled for when modeling retweet activity (below).

## Self efficacy

Efficacy information [9,22] instructs individuals about how they can protect themselves against a threat. Efficacy keywords express actions that an individual can take to protect themselves from being exposed to coronavirus or exposing others to coronavirus such as staying home, social distancing, washing hands, or wearing a mask.

## Collective efficacy

Collective efficacy refers to the capacity to achieve an intended effect [23] such as limiting population exposure and reducing virus transmission in order to protect vulnerable populations and critical resources. Collective efficacy includes phrases such as slow the spread, good neighbors, united, solidarity, flatten the curve, and stay home save lives.

## Technical information

Technical information includes keywords describing the mechanisms of how the virus spreads [9]. Example words and phrases include droplet, transmission, incubation, person-to-person, and large gatherings.

## Official response

Official response includes keywords about governmental responses and how to access information [9]. Example words and phrases include policy, declaration, proclamation, restrictions, and surveillance as well as briefing, resource guide, and press conference.

## Resilience

Resilience keywords express thanks and appreciation to the work of community members who provide essential services and supplies during the response [3]. These words include hero, salute, champion, honor, grateful, and thank.

## Closures and openings

Closures and openings keywords indicate the suspension of services, activities, and facilities [3]. Keywords include visitation, lock down, mandatory, recreation, gatherings, parade, ceremony, as well as salon, spa, nightclub, bar, NBA, and NCAA.

## Additional keyword categories

Thematic categories that emerged included keywords about the **primary threat**, including COVID19, coronavirus, and nCoV-SARS and **secondary threats** that developed over the course of the event. This includes topics such as mental health, substance abuse, child abuse, domestic violence, food and housing insecurity, blood shortages, scams and misinformation, unemployment, stigma and discrimination, schools, price gouging, medical supply and equipment shortages, essential services, and field hospitals. We also include keywords for **information sharing**, such as webinar, resource guide, hotline, and briefing, identifying places, times, or strategies to obtain additional information about the ongoing pandemic.

And finally, because our data include tweets from the month of February, when there was considerably less attention on the pandemic, we found a number of keywords that helped to identify clearly **off topic tweets**. These keywords were primarily associated with current events, such as Super Bowl, puppy bowl, Valentines Day, Black History Month, groundhog day, State of the Union, and census. We also found a number of campaigns, frequently associated with hashtags, such as #goredforwomenday and #cancerawareness.

## Coding message features

We used regular expressions to code messages for content, structure, and style (see Table 1). Codes were distinctive but not mutually exclusive. Messages could exhibit characteristics for each stylistic and structural feature as well as multiple content keywords.

Messages were coded for the thirteen content codes described previously, and nine structural features. Structural features are coded for presence or absence in a given message, which included a video, an image, a hashtag, a hyperlink (URL), a mention (@username), or the reply function. We also include "quote tweets" as a type of message structure. Quote tweets are tweets that repost the original tweet with additional comment. To help understand the sentence style of a given message we also coded for punctuation, demonstrated by the presence of a question mark or an exclamation point.

## Data analysis

**Engagement.**   First, to assess whether message content and structure varied between the two time periods of interest (before and after the national emergency declaration), we used chi-square tests to examine the association for message content and message structural features and whether the message was published before and after March 13, 2020. Since chi-square tests can only inform us whether an association between two variables exists, to understand the magnitude and direction of their relationship we calculated the log odds ratios and odds ratios. Odds ratios are more naturally interpreted on log scale due to their being multiplicative: null effects are mapped 0 and effect sizes can be compared symmetrically to one another. Message features that exhibit positive log odds ratios indicate that there is an increase use during the post-declaration period, while a negative log odds-ratio indicates a lower use during the post-declaration period. Each message feature's Chi-square statistic, log odds ratio, and raw odds ratios were calculated separately using the R Statistical Programming Language.

**Message retransmission.**   The other outcome of interest for this study is the propensity for messages to be retransmitted. Retransmission was operationalized as the total number of times that a given message was retweeted or passed on–with the core question for our analyses being *how do risk communication strategies affect message retransmission relative to the declaration of a national emergency*? To estimate the contribution of multiple mechanisms to information retransmission, we perform a negative binomial regression of the retweet count on predictors related to message style, content, and posting context. We parameterize our model as.

$$\log \mathbf{E}Y_i = \beta^T X_i, Y_i \sim \text{NegBin}(\mathbf{E}Y_i, \varphi)$$

where $Y_i$ is the retweet count for the $i$th message (with expectation $\mathbf{E}Y_i$ and covariate vector $X_i$), $\beta$ is a vector of regression coefficients, and $\varphi$ is the dispersion parameter for the negative binomial distribution. As this implies, a one-unit change in $X_i$ under this model is associated with a change of $\beta$ in the log expected retweet count, or equivalently a multiplicative change in the expected retweet count by a factor of $\exp(\beta)$. Use of the negative binomial likelihood allows us to account for the high level of overdispersion in the data, arising from the well-known tendency of some messages to "go viral" while other, similarly situated messages receive less attention.

As has been found in prior work on retransmission in the domain of Twitter [3], the majority of messages published by official agencies do not get retweeted, with 65% of messages in our sample being passed on less than eight times. However, there are some messages that are retweeted at much higher rates, which results in a highly skewed distribution (Mean = 59, Median = 3, Mode = 0 [26% of all messages], Standard Deviation = 685.27, Skewness = 44.63, Kurtosis = 2954.36). The marginal distribution of retweet frequency is shown in Fig 2. As noted above, we account for this highly skewed distribution by employing a negative binomial model [24]; models were fit using the MASS and glmmADMB packages in the R statistical programming language [25,26].

We also control for various contextual factors that are unrelated to the content and structural features of a given message using fixed effects. These factors include seasonal effects including the month, the day of the week, and the time of the day a message was sent. (The large volume of available messages allows us to reliably estimate daily cycles at hourly resolution, as can be seen from the standard errors in Table 3.) We also control for the type of account sending the message: public health organizations, governors, mayors, state emergency management agencies, and local emergency management agency accounts.

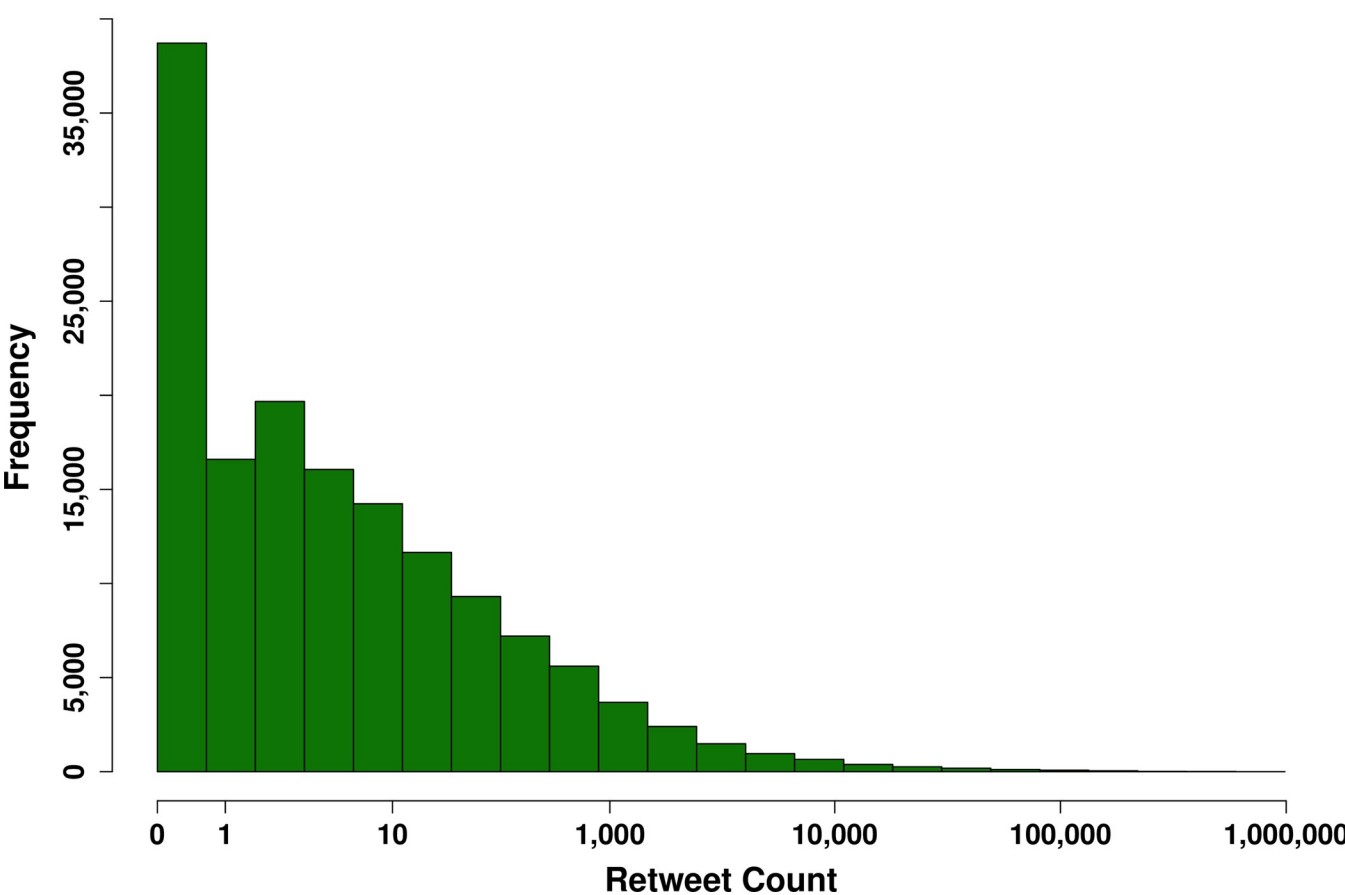

**Fig 2. Marginal distribution of retweet frequency over all messages.** As is typical for online messages, most messages receive few retweets. A small fraction of messages, however, are reshared by extremely large numbers of users.

## Results

Messages most often contained content about official response ($n$ = 28,677; 19%), individual efficacy ($n$ = 28, 324; 19%), and surveillance ($n$ = 27,465, 18%). In regard to message structure presented in tweets, just over 49% ($n$ = 73,091) of tweets included URLs, 45% included a hashtag ($n$ = 67,876) and 45% included a photo ($n$ = 66,830).

The chi square analyses indicate significant differences in message content and message structure in relation to the pre- and post- emergency declaration periods. (odd ratios reported in Table 2). First, the inclusion of some content categories and structural features *decrease* following the emergency declaration. For example, **off-topic** content decreased by 63.5%, while content about **technical information** decreased by 18.6%, content about **susceptibility** decreased by 17.5%, and content about **closures and openings** decreased by 17% (Fig 3).

We also find that messages containing an **exclamation point** or a **question mark** decrease by 48.6% and 27% respectively following the emergency declaration.

**Table 2. Chi square and odds ratios for message content, features, and account type.**

|  | $X^2$ | Odds | CI Lower | CI Upper | Sig |
|---|---|---|---|---|---|
| Off Topic | 3016.3 | 0.365 | 0.351 | 0.379 | *** |
| Exclamation Point | 2045.2 | 0.514 | 0.499 | 0.529 | *** |
| Question Mark | 216.6 | 0.728 | 0.698 | 0.760 | *** |
| Incl. Image | 709.0 | 0.738 | 0.722 | 0.755 | *** |
| Local EM Account | 202.6 | 0.781 | 0.755 | 0.809 | *** |
| Technical Info. | 71.9 | 0.813 | 0.775 | 0.853 | *** |
| Susceptibility | 99.1 | 0.825 | 0.794 | 0.857 | *** |
| Incl. Mentions | 252.2 | 0.827 | 0.808 | 0.847 | *** |
| Closure/Openings | 125.1 | 0.828 | 0.801 | 0.856 | *** |
| Incl. Hashtag | 141.0 | 0.873 | 0.854 | 0.893 | *** |
| Incl. URL | 83.7 | 0.901 | 0.881 | 0.921 | *** |
| State EM Account | 6.8 | 0.939 | 0.895 | 0.984 | ** |
| Public Health Account | 3.6 | 0.979 | 0.957 | 1.001 | NS |
| Resilience | 0.1 | 0.989 | 0.932 | 1.049 | NS |
| Incl. Video | 0.0 | 1.000 | 0.945 | 1.057 | NS |
| Mayor Account | 6.2 | 1.037 | 1.008 | 1.066 | * |
| Official Response | 6.4 | 1.037 | 1.008 | 1.067 | * |
| Incl. Quote | 26.7 | 1.107 | 1.065 | 1.150 | *** |
| Governor Account | 183.2 | 1.236 | 1.199 | 1.275 | *** |
| Primary Threat | 146.5 | 1.248 | 1.204 | 1.294 | *** |
| Symptoms | 39.7 | 1.268 | 1.178 | 1.365 | *** |
| Reply | 435.1 | 1.326 | 1.291 | 1.362 | *** |
| Actions/Efficacy | 596.5 | 1.453 | 1.410 | 1.498 | *** |
| Surveillance | 662.8 | 1.493 | 1.448 | 1.540 | *** |
| Secondary Impacts | 548.2 | 1.497 | 1.447 | 1.548 | *** |
| Info. Sharing | 595.8 | 1.609 | 1.549 | 1.673 | *** |
| Collective Appeals | 763.5 | 1.788 | 1.715 | 1.865 | *** |

NS $p \geq 0.05$,

* $p < 0.05$,

** $p < 0.01$,

*** $p < 0.001$

Table 3. Negative binomial regression model predicting message passing.

|  | Estimate | Exp. Beta | Std. Error | Sig. |
|---|---|---|---|---|
| Intercept | -4.754 | 0.009 | 0.055 | *** |
| *Account Properties* |  |  |  |  |
| Governor Account | 1.00 | 2.719 | 0.013 | *** |
| Log Follower Count | 0.760 | 2.138 | 0.002 | *** |
| Mayor Account | 0.135 | 1.145 | 0.011 | *** |
| Log(+1) Friends Count | -0.073 | 0.930 | 0.004 | *** |
| State EM Account | -0.183 | 0.833 | 0.017 | *** |
| Local EM Account | -0.668 | 0.513 | 0.014 | *** |
| *Microstructural Properties* |  |  |  |  |
| Incl. Video | 0.489 | 1.631 | 0.020 | *** |
| Incl. Hashtag | 0.121 | 1.128 | 0.008 | *** |
| Incl. Image | 0.059 | 1.061 | 0.009 | *** |
| Incl. Quote | -0.072 | 0.931 | 0.015 | *** |
| Incl. Question Mark(?) | -0.074 | 0.929 | 0.016 | *** |
| Incl. Mention | -0.261 | 0.770 | 0.008 | *** |
| Incl. Exclamation(!) | -0.264 | 0.768 | 0.011 | *** |
| Incl. URL | -0.349 | 0.705 | 0.009 | *** |
| Reply | -1.677 | 0.187 | 0.010 | *** |
| *Message Content* |  |  |  |  |
| Surveillance | 0.339 | 1.404 | 0.010 | *** |
| Technical Info. | 0.266 | 1.305 | 0.017 | *** |
| Actions/Efficacy | 0.246 | 1.279 | 0.010 | *** |
| Symptoms | 0.238 | 1.269 | 0.024 | *** |
| Primary Threat | 0.195 | 1.216 | 0.012 | *** |
| Secondary Impacts | 0.183 | 1.201 | 0.011 | *** |
| Official Responses | 0.174 | 1.190 | 0.010 | *** |
| Collective Appeals | 0.125 | 1.133 | 0.013 | *** |
| Closures/Openings | 0.116 | 1.123 | 0.012 | *** |
| Resilience | 0.066 | 1.068 | 0.020 | ** |
| Susceptibility | 0.046 | 1.047 | 0.013 | *** |
| Off Topic | -0.008 | 0.992 | 0.014 | NS |
| Info. Sharing | -0.118 | 0.889 | 0.013 | *** |
| *Period Effects–National Emergency Declaration Period* |  |  |  |  |
| Post-Declaration | 0.369 | 1.446 | 0.014 | *** |
| *Period Effects–Month* |  |  |  |  |
| March | 0.931 | 2.537 | 0.015 | *** |
| April | 0.460 | 1.584 | 0.018 | *** |
| *Period Effects–Time of Day* |  |  |  |  |
| 12 am UTC | -0.704 | 0.495 | 0.048 | *** |
| 1 am UTC | -0.511 | 0.600 | 0.049 | *** |
| 2 am UTC | -0.071 | 0.932 | 0.051 | NS |
| 3 am UTC | -0.009 | 0.991 | 0.054 | NS |
| 5 am UTC | -0.357 | 0.700 | 0.075 | *** |
| 6 am UTC | -0.159 | 0.853 | 0.095 | NS |
| 7 am UTC | -0.571 | 0.565 | 0.122 | *** |
| 8 am UTC | -0.423 | 0.655 | 0.099 | *** |
| 9 am UTC | -0.917 | 0.400 | 0.093 | *** |

(*Continued*)

**Table 3.** (Continued)

| | Estimate | Exp. Beta | Std. Error | Sig. |
|---|---|---|---|---|
| 10 am UTC | -0.868 | 0.420 | 0.080 | *** |
| 11 am UTC | -0.875 | 0.417 | 0.059 | *** |
| 12 pm UTC | -0.790 | 0.454 | 0.050 | *** |
| 1 pm UTC | -0.844 | 0.430 | 0.048 | *** |
| 2 pm UTC | -0.681 | 0.506 | 0.047 | *** |
| 3 pm UTC | -0.822 | 0.440 | 0.046 | *** |
| 4 pm UTC | -0.747 | 0.474 | 0.046 | *** |
| 5 pm UTC | -0.796 | 0.451 | 0.046 | *** |
| 6 pm UTC | -0.905 | 0.404 | 0.046 | *** |
| 7 pm UTC | -0.715 | 0.489 | 0.046 | *** |
| 8 pm UTC | -0.834 | 0.434 | 0.046 | *** |
| 9 pm UTC | -0.791 | 0.454 | 0.046 | *** |
| 10 pm UTC | -0.787 | 0.455 | 0.047 | *** |
| 11 pm UTC | -0.762 | 0.467 | 0.047 | *** |
| *Period Effects–Day of Week* | | | | |
| Sunday | 0.262 | 1.299 | 0.016 | *** |
| Monday | 0.146 | 1.157 | 0.013 | *** |
| Tuesday | 0.044 | 1.045 | 0.013 | *** |
| Thursday | 0.077 | 1.080 | 0.013 | *** |
| Friday | 0.074 | 1.076 | 0.013 | *** |
| Saturday | 0.192 | 1.212 | 0.015 | *** |

Observations: 149335; AIC: 966230,

Log-Likelihood: -483053; Dispersion Parameter: 0.542; Std. Error: 0.002

* $p < 0.05$,

** $p < 0.01$,

*** $p < 0.001$

The use of various message structural features also decrease post-emergency declaration. For example, we find a 26% decrease of the inclusion of **images**, a 17% decrease in **mentions**, a 12% decrease in the use of **hashtags**, and a 10% decrease in the inclusion of **urls** following the emergency declaration.

In contrast, the inclusion of some message contents and structures *increase* post-emergency declaration. For example, content about **collective efficacy** increases by 78%, while content about **information sharing** increases by 61%, content about **secondary impacts** increases by 50%, content about **surveillance** increases by 50%, and content about **individual efficacy** increases by 45%. Additional messaging content changes post-declaration include increases in content about **official response activities** (3.7%), the **primary threat** (25%), and **symptoms** (27%).

There are also several message structural features that increase in frequency. For example, **quote tweets** increase by 10% and **direct replies** to other users increase by 32.5%.

We also find that there are a few message features that remain fairly consistent regardless of the period during which they were used. For example, **video attachments** and content about **resilience** are constant in messages.

And finally, we find that organizations varied their frequency of tweeting pre- and post-emergency declaration. Following the declaration, messages posted by **local emergency management** accounts decreased by 22% while messages by **state emergency management**

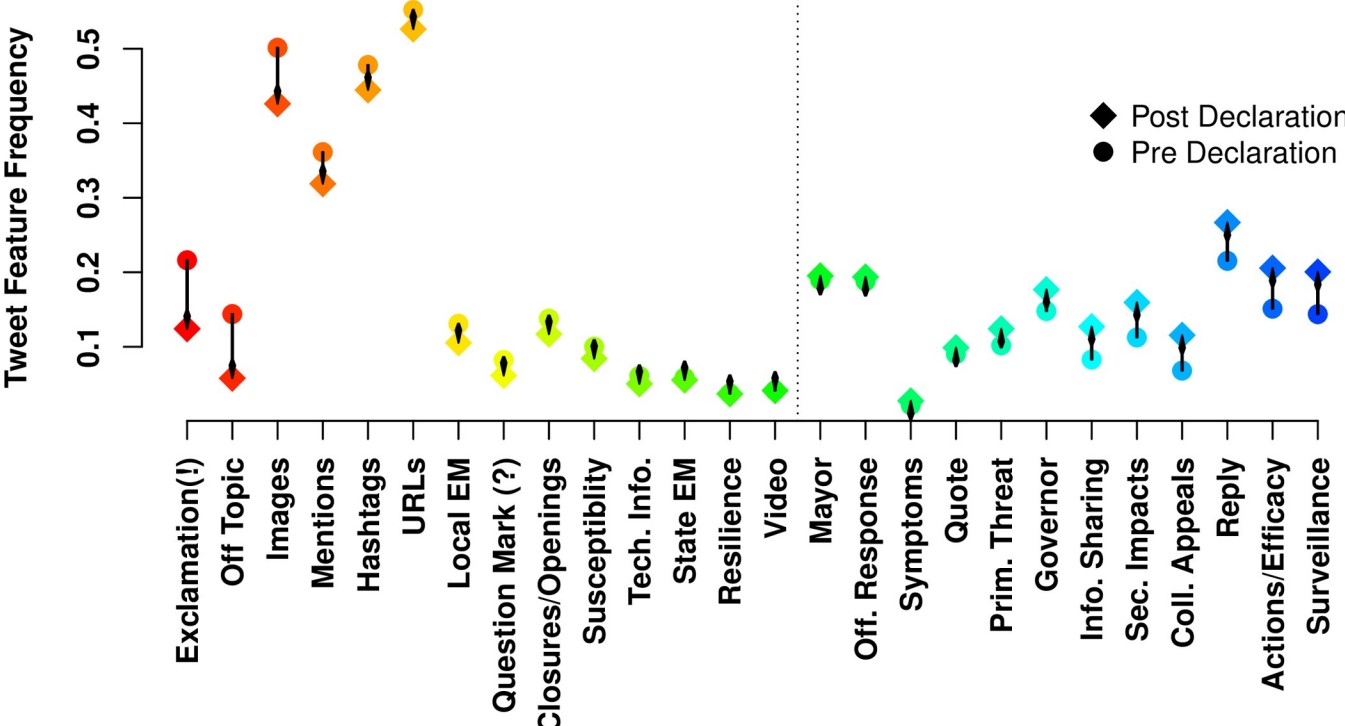

**Fig 3. Frequency of message feature use, pre- versus post- emergency declaration.** Vertical axis indicates fraction of tweets containing each feature (horizontal axis), with features sorted by pre/post change; features to the left of the dotted line decline in prevalence post-declaration, while those to the right increase. Overall, we see a marked reduction in informal and off-topic modes of communication, with an enhanced emphasis on actionable content.

accounts decreased by 6%. In contrast, **mayors accounts** increased their messaging by 3.6% and **governors accounts** increased the rate of their messaging by 23.6%. There were no significant differences in frequency of messages posted by **public health** organizations before or after the emergency declaration.

## Message retransmission

The reported model (see Table 2) indicates that the retweeting of risk communication messages during the first three months of the COVID-19 pandemic is influenced by message content, message features, organizational type of the account, the number of followers for an account, and the time/day at which a message is sent.

## Message content

Message contents, structure, and style jointly influence message retransmission. Content features that most strongly influenced message retransmission pertained to information characterizing the impacts of the virus, its spread, and actions individuals can take to protect themselves (See Fig 4). Messages that include **surveillance** content were passed on 40% more often than messages that did not include that content. Messages that include **technical information** were passed on 30% more often; those containing content about **efficacy** were passed on 28% more often; and those containing content about **symptoms** were passed on 27% more often than messages that did not include those contents. Messages including content naming the **primary threat**, were passed on 21.5% more, while **secondary threats** from the virus were passed on 20% more. Messages containing content about **official responses** to the pandemic

**Message Keyword Categories**

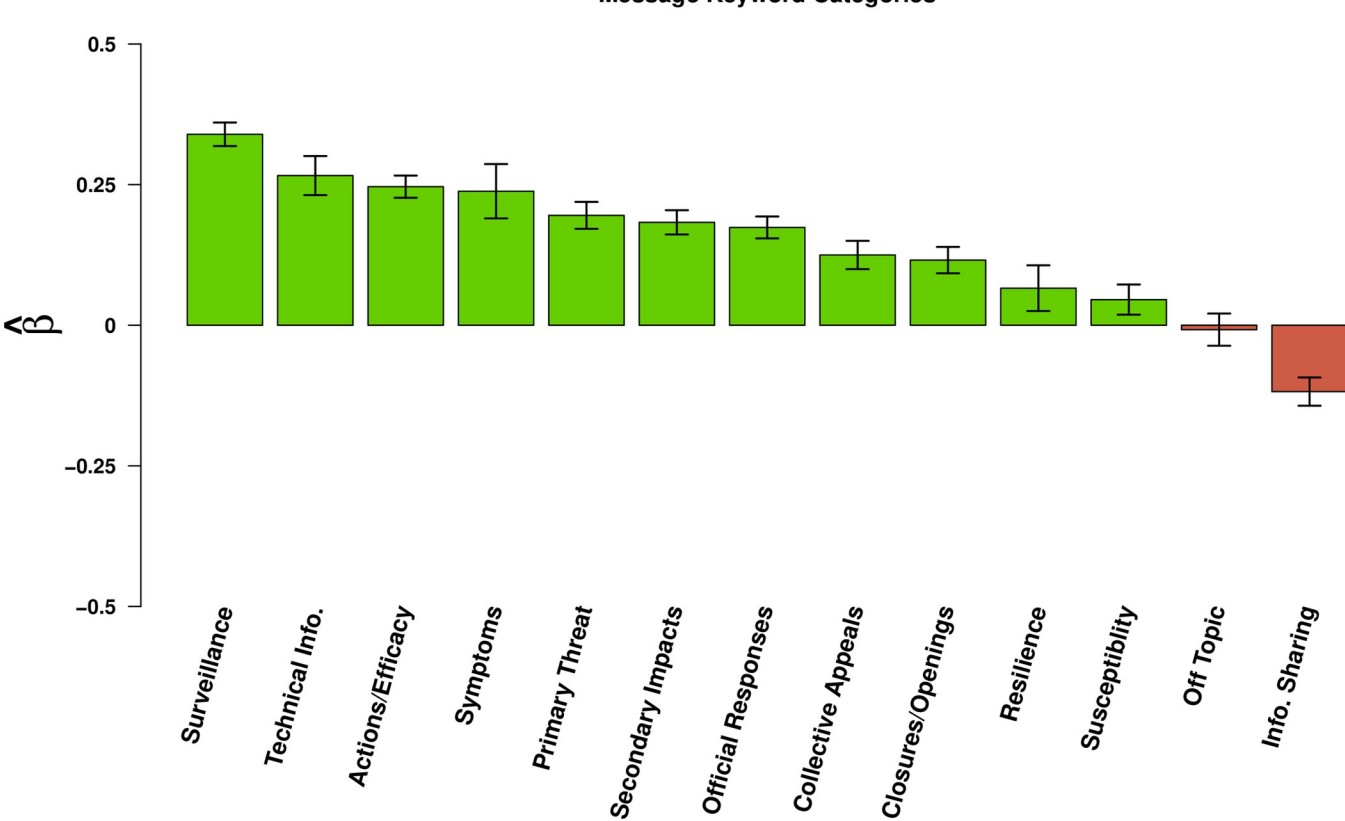

**Fig 4. Effects of message content on retransmission.** Bars indicate effects of content covariates (horizontal axis) on log expected retweet count (see Table 2); whiskers indicate 95% confidence intervals. A wide range of COVID-19 related message content enhances retransmission, with technical information, information related to disease surveillance and symptoms, and actions that can be taken to prevent infection being among the most important predictors. By contrast, content identifying sources of follow-up information (information sharing) tends to suppress retransmission.

were shared 19% more often, while messages with content about **collective efficacy** were passed on 12.5% more and **closures and openings** were passed on 12% more than those that did not contain that content. Messages containing content about **resilience** and **susceptibility** had the smallest effect on message retransmission with increases of 6.8% and 4.6% respectively. While most message contents have a positive effect on message retransmission, we find that messages containing content about **information sharing** decrease the likelihood of being shared by 11%.

## Message structural features and sentence style

We also find that some structural features increase message retransmission (See Fig 5). The inclusion of **media**, both video and images increase message passing. Messages that include **videos** are retransmitted 63% more often and messages containing **photos** or other images are passed on 27% more often than those that do not, holding all other features constant. We also find that the use of **hashtags** increase message passing by about 12% more than those that do not include a hashtag.

Importantly, there are several structural features that decreased message retransmission. For example, '**quote tweets**,' that is, posting original content while quoting another user's message leads to a 7% decrease in message passing. We also find that **mentioning** another account or **directly replying** to another user decrease message passing by 23% and 82% respectively.

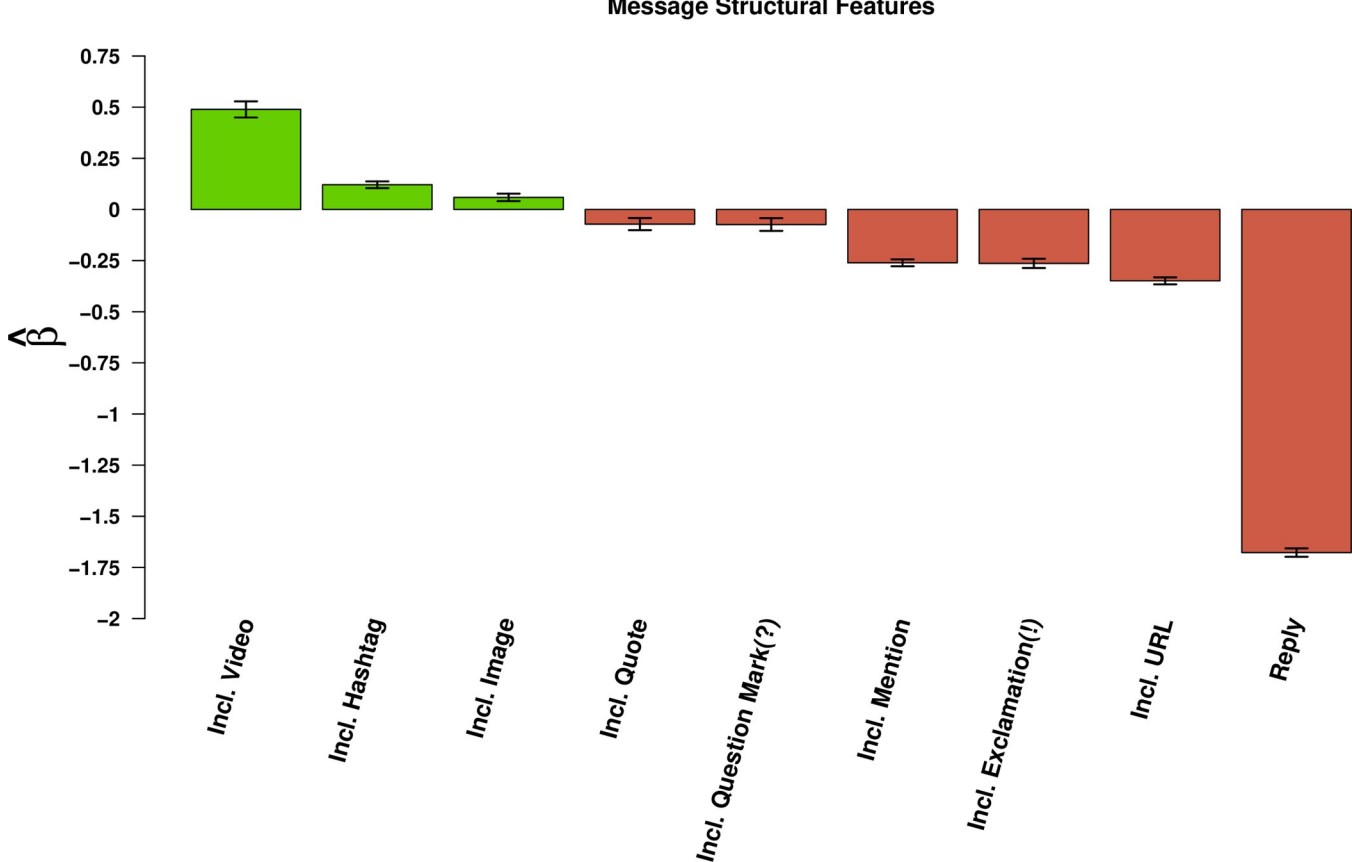

**Fig 5. Effects of message structure on retransmission.** Bars indicate effects of content covariates (horizontal axis) on log expected retweet count (see Table 2); whiskers indicate 95% confidence intervals. Videos substantially enhance retransmission, while photos and hashtags have much more modest effects. Informal language (e.g., exclamatory and interrogative text), audience-narrowing features (e.g., mentions and replies), and URLs tend to suppress retransmission.

And finally, we find that messages containing **weblinks** (URL) decrease the likelihood of retransmission by 30% holding all other features constant.

We also find that sentence style affect message passing. Messages that include **exclamation marks** were 23% less likely to be shared; messages including **question marks** were 7% less likely to be shared. This is distinct from some other settings, in which such saliency-enhancing features increase retransmission rates.

## Period, organization, and follower effects

The date at which the message was sent also makes a difference in message retransmission. For example we find that messages published after the national emergency declaration on March 13[th] were passed on 44% more frequently than those sent before the declaration. We also find that, when compared to the reference category of February, messages posted in the month of March were retransmitted 153% more often; those posted in April had an increase of 58%.

The account organizational type also had an impact on message passing. When compared to the reference category of the public health accounts, we find that messages posted from governors accounts were retransmitted 172% more frequently, while those from mayors accounts were shared 14% more often. In contrast, we find that messages posted by emergency management accounts decreased message retransmission. Those posted by state emergency

management saw a 17% decrease; those posted by local emergency management saw a decrease of 49%.

Finally, we find that the number of followers has a significant effect on message retransmission. We find that a one unit increase in the log count of an account's followers coincides with a 113% increase in message passing. Period, organization and follower effects are summarized in Fig 6.

## Discussion

This analysis has investigated two aspects of messaging by official accounts from February 2020, before widespread acknowledgment or impact of COVID-19 across the U.S. to the end of April 2020, when health officials began to emphasize the use of masks and face coverings to limit the spread of the virus and more than 60,000 persons had died. We first examined the volume of messages by account type and the inclusion of message features, including content, structure, and style, and how those features differed pre- and post- emergency declaration. Next we modeled the use of those message features to determine how their inclusion positively or negatively affected message retransmission. Here, we discuss how message content and features employed by official accounts interact with message retransmission.

One of the strongest contributors to message retransmission is the *message structures* that are technological affordances of the Twitter platform. In the case of the pandemic, we find that

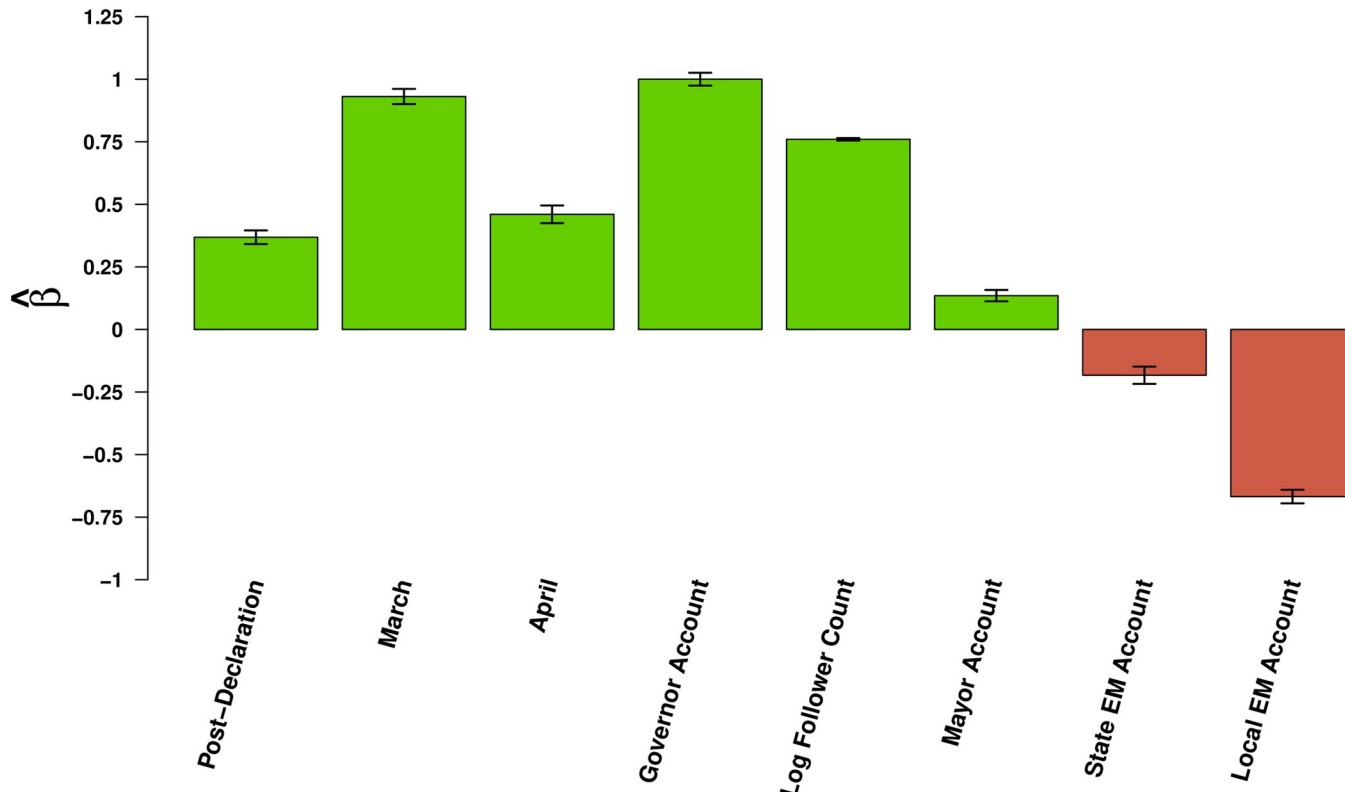

**Fig 6. Effects of time period and account type on retransmission.** Bars indicate effects of content covariates (horizontal axis) on log expected retweet count (see Table 2); whiskers indicate 95% confidence intervals. Relative to the pre-declaration period, retransmission rates are higher in the post-declaration period; base retransmission rates increased further in March, 2020, while falling back somewhat in April. Relative to public health agencies, elected officials are frequently retweeted, while state and local emergency management accounts see less retransmission. As in other settings, follower counts are also an important predictor of retransmission.

the inclusion of **media**, both videos and photos, lead to significantly greater message sharing. Prior research has shown that media is an important feature that draws attention to messages, helping them to stand out in a sea of text [9]. In this research, while there is little change in the frequency of including videos with tweets, there was a decrease in the use of images post-declaration.

The inclusion of **hashtags** also increase message passing. Prior research has shown that hashtags serve as strategy for organizing informational topics [27] and helping groups to form and exchange information [28], and they become especially important for coordinating localized information [15]. Here we find that while hashtag use decreases post-declaration, it remains an important predictor of message passing. It is possible that organizations were actively using hashtags to communicate about campaigns or other planned events in the pre-declaration period in a coordinated fashion among organizations, which then declined when attention was turned to the pandemic [29].

Prior research has also shown that structural features also lead to a decrease in message retransmission. In particular, the inclusion of **directed messages**, in the form of replies, consistently and negatively affects message retransmission under conditions of imminent threat [3] and during emerging infectious disease. Directed messages serve as a strategy for engagement between an organization and a single message sender; therefore it is unsurprising that these messages would not be retransmitted broadly. In this case, we find that the use of directed messages significantly increased post-emergency declaration. This is similar to the finding from prior research on the use of Twitter during periods of increased threat by National Weather Service Weather Forecast Offices (WFOs). Under conditions of greater uncertainty, WFOs increased their direct replies to users; in this research we find that public agencies also increase their replies, suggesting an effort to engage with persons in an ongoing manner [15]. While directed messages do not lead to increased message retransmission, they may serve to increase trust in the organization due to its responsiveness [5].

Another structural feature that has been found to negatively affect message passing is the inclusion of a **URL** [3]. Here we find a decline in the use of this messaging strategy in the post-declaration period. However, this consistently negative effect on message passing suggests that it may be more effective to deliver additional information by attaching content as an image if the goal is to increase message passing among online individuals.

We also find that **quote tweets** also negatively affect message retransmission. While this structural feature increased in use post-declaration, its use leads to a decrease in message passing. Such a finding suggests that attaching commentary to a previously constructed message doesn't aid in the diffusion of that content. If the goal of the sending organization is to increase the reach of an original message, it may be better to simply retweet it.

Message *content* also affects retransmission. Prior research found that messages that contain information about the hazard and its impact [3] or its severity [9] will increase message passing. Here we find that tweets containing content about **surveillance**, referencing the extent of those infected, hospitalized, dead, or recovered, and tweets containing **technical information**, that is how the hazard is spread among populations, both positively and significantly affect message retransmission. Such attention to the hazard and its ongoing impact is not unexpected given the initial uncertainty about how it spread and the severity of the impacts on different populations. As multiple versions of epidemiological models were developed and shared, attempts to predict the infection rate with and without mitigation could only be verified by the posting of daily surveillance numbers that report deaths and any evidence that the curve was flattening. Messages containing surveillance content increased by more than 50% post-declaration.

Additionally, protective action guidance, in the form of a warning or advice [3] or efficacy [9], has previously been shown to increase message retransmission as individuals pass along information about what can be done to limit exposure to a threat and to protect themselves from harm. Here we find that messages with content about **individual efficacy** were passed on 28% more often than those without efficacy content. However, in the case of COVID-19, we also find that officials posted content about **collective efficacy**, that is content about social actions that taken together as a broad population are necessary to reduce the spread of the disease to the most vulnerable. Messages exhorting people to "flatten the curve," "stop the spread," and "stay home save lives", increased by 78% post-emergency declaration and had a positive effect on message retransmission. In contrast, we find that **Information sharing** has a negative effect of retransmission. Messages that include information sharing decreases retransmission; however, this content increased post emergency declaration by 61%.

Previous research also showed that the *style* of a message, in the form of exclamatory or interrogative sentences, has an effect on message retransmission. In particular, messages with exclamation points frequently draw attention to the text and suggest urgency on the part of the sender [3]. Here we see not only a decrease in the use of exclamatory sentences post-declaration, we also find the use of **exclamation marks** decreased retransmission. It is possible that in this case of a slow moving virus where orders were released and widespread measures were implemented gradually, perhaps very little was so urgent to warrant an exclamation point. Equally plausible, however, is the observation that exclamatory and interrogative language is relatively *informal*, and potentially viewed as less authoritative than declarative language. In an environment in which members of the public are seeking authoritative guidance regarding a threat that they already perceive, trading authority for salience may be a poor bargain. It is possible that this balance may shift in the future, if gaining and maintaining attention becomes a challenge due to response fatigue.

We also find that *timing* of the message has a strong effect on message retransmission. Previous research has found an increase in messages posted leading up to and during imminent threat events by both members of the public [6] as well as response organizations [2] serves as a measure of salience about the event (how many people are paying attention) and also demonstrating the usefulness of social media as a platform for collective communication. We also find that the *type of organization* posting messages affects retransmission. Perhaps it is expected that accounts representing individuals who are responsible for decision making, including monitoring the health of its local citizenry, declaring shelter in place orders, and placing restrictions on businesses and other organizations, would also receive a great deal of attention during a widespread pandemic. Governors and mayors not only posted messages more often post-declaration, their messages were also retweeted more frequently than other organization types. In contrast, emergency management accounts at both the state and local level decreased message frequency post-declaration.

And finally, *follower numbers* matter. Due to the effects of network diffusion, messages posted by accounts with more followers will not only receive greater immediate exposure but are likely to receive more retweets overall (thus exposing them to yet more users). This effect is a consistent finding in work on organizational communication on Twitter, emphasizing the importance of building a following over the long term.

## Limitations

This research provides insight into the content and style features employed by official accounts pre- and post- emergency declaration and models what features increase or decrease message retransmission. From this, we are able to draw conclusions about how to improve message

design to increase diffusion of official accounts during an emerging public health crisis. We do not know how these results apply to other types of accounts, such as media or other entities. Additionally, we note that all social media efforts will be hampered by the algorithms deployed by the platforms themselves, which prioritize some accounts and messages to the detriment of others. We also cannot know the effects these messages actually have on behaviors offline. While retransmission demonstrates some form of interaction with content leading to a decision to share the message more broadly, we don't know what occurs beyond choosing to amplify the message. And, finally, while the findings on message retransmission remain fairly consistent across hazards and across account types, changing circumstances may lead to different patterns in the future. Usage of these findings to guide practice should bear in mind the potential for retransmission predictors to evolve as the pandemic unfolds.

## Conclusions

A central feature of the COVID-19 pandemic is the need to respond to rapidly changing circumstances, due both to the changes in the state of public health knowledge (e.g. on the efficacy of masks for personal protection) and the evolving political and economic situation (e.g. distancing regulations and resistance thereto). Pandemics are, ultimately, disasters, and the critical role of improvisation that is central to effective disaster response is also inescapable here: it is unlikely in the case of COVID-19 that conditions can be expected to fully stabilize in the near future, and risk communicators within official agencies need to establish processes to continuously re-assess and re-evaluate messaging practices in light of changing events (where possible, anticipating messaging that could be used if particular future conditions were to obtain). Our work has shown that, in this uncertain environment, the public has been attentive to—and likely to retransmit—a wide range of practical information regarding the health impacts of COVID-19, protective action measures, and the progress of the pandemic itself. At the same time, we have also found that some saliency-enhancing tactics useful in other disasters (such as sentence styles that use exclamatory and interrogative punctuation) have been counterproductive in the COVID-19 pandemic. Employing these factors when crafting messages may help public agencies in reaching the public in later stages of the COVID-19 pandemic, or in the next public health emergency to arise.

## Supporting information

**S1 Data.**
(XLSX)

## Author Contributions

**Conceptualization:** Jeannette Sutton, Carter T. Butts.

**Data curation:** Scott L. Renshaw.

**Formal analysis:** Jeannette Sutton, Scott L. Renshaw.

**Funding acquisition:** Jeannette Sutton, Carter T. Butts.

**Investigation:** Jeannette Sutton, Scott L. Renshaw.

**Methodology:** Carter T. Butts.

**Project administration:** Jeannette Sutton.

**Supervision:** Jeannette Sutton, Carter T. Butts.

**Visualization:** Scott L. Renshaw.

**Writing – original draft:** Jeannette Sutton, Scott L. Renshaw.

**Writing – review & editing:** Carter T. Butts.

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
