## [Decision Letter · Decision Letter 0]

10 Jul 2020

PONE-D-20-17359

COVID-19: Retransmission of Official Communications in an Emerging Pandemic

PLOS ONE

Dear Dr. Sutton,

Thank you for submitting your manuscript to PLOS ONE. After careful consideration, we feel that it has merit but does not fully meet PLOS ONE’s publication criteria as it currently stands. Therefore, we invite you to submit a revised version of the manuscript that addresses the points raised during the review process.

The reviewers both asked for reasonable minor revisions before considering acceptance, please address their comments in full.

We look forward to receiving your revised manuscript.

Kind regards,

Christopher M. Danforth

Academic Editor

PLOS ONE

Journal Requirements:

2.We note that Table 1 in your submission contain copyrighted images. All PLOS content is published under the Creative Commons Attribution License (CC BY 4.0), which means that the manuscript, images, and Supporting Information files will be freely available online, and any third party is permitted to access, download, copy, distribute, and use these materials in any way, even commercially, with proper attribution. For more information, see our copyright guidelines: http://journals.plos.org/plosone/s/licenses-and-copyright.

a)    You may seek permission from the original copyright holder of the figures in Table 1 to publish the content specifically under the CC BY 4.0 license.

4.We note that you have stated that you will provide repository information for your data at acceptance. Should your manuscript be accepted for publication, we will hold it until you provide the relevant accession numbers or DOIs necessary to access your data. If you wish to make changes to your Data Availability statement, please describe these changes in your cover letter and we will update your Data Availability statement to reflect the information you provide.

Reviewers' comments:

Reviewer's Responses to Questions

**Comments to the Author**

1. Is the manuscript technically sound, and do the data support the conclusions?

Reviewer #1: Yes

Reviewer #2: Yes

2. Has the statistical analysis been performed appropriately and rigorously? 

Reviewer #1: Yes

Reviewer #2: Yes

3. Have the authors made all data underlying the findings in their manuscript fully available?

Reviewer #1: No

Reviewer #2: Yes

4. Is the manuscript presented in an intelligible fashion and written in standard English?

Reviewer #1: No

Reviewer #2: Yes

5. Review Comments to the Author

Reviewer #1: This paper analyzes the retransmission of the messages on Twitter that come from the official sources and that are related to COVID-19. The authors build the set of features for the tweets and examine the importance of each feature to message retransmission.

The message dissemination is an important topic, and has been studied extensively in the last few years. It is important however, to have a study that is focused on this topic during the times of crisis such as global pandemic.

The paper is neat, easy to understand and timely. I was enjoying reading it.

I find the most important contribution of the paper to be the topic lexicon, that will be potentially very useful for other researcher dealing with the communication in times of crisis. Another important result is the analysis of the most effective message structures that will drive the retransmission. The features are interpretable and the results could be used to improve the official communication channels.

The results of the model are not surprising, and I miss seeing the comparison to some baseline. A possible baseline could be the analysis of some other accounts during the crisis. Maybe the results are applicable elsewhere, and not only for the official accounts. This is not particularly important, but can be useful in the future work.

A suggestion - When measuring period effects, it is more useful to divide the time of the day to morning, afternoon, night... Splitting it on the hourly basis is too fine-grained and can result in smaller sample sizes and the lower confidences.

Nice paper. I would like to see it published.

Reviewer #2: - I recommend including a visualization of the retweet distribution to demonstrate its skew. Using a log-log scale would help with this.

- I'd like to see the formula for the negative binomial regression model written somewhere.

- Typo: "retweeing", page 22

- In the first few paragraphs of the discussion, the several references to prior research make it somewhat difficult to tell which statements are being attributed to your original work. Some simple phrasing (eg. "we find that...") can help with this.

- In the conclusion, it feels like a reach to attribute lower likelihoods to exclamatory and interrogative "language", given that this part of the analysis was entirely based on punctuation.

6. PLOS authors have the option to publish the peer review history of their article (what does this mean?). If published, this will include your full peer review and any attached files.

Reviewer #1: No

Reviewer #2: No

---

## [Author Response · Author response to Decision Letter 0]

14 Aug 2020

We have uploaded a document with a response to reviewers.

---

## [Editor Report · Decision Letter 1]

19 Aug 2020

COVID-19 Retransmission of official communications in an emerging pandemic.

PONE-D-20-17359R1

Dear Dr. Sutton,

We’re pleased to inform you that your manuscript has been judged scientifically suitable for publication and will be formally accepted for publication once it meets all outstanding technical requirements.

Kind regards,

Christopher M. Danforth

Academic Editor

PLOS ONE
---

## [Editor Report · Acceptance letter]

24 Aug 2020

PONE-D-20-17359R1 

COVID-19:
Retransmission of official communications in an emerging pandemic.  

Dear Dr. Sutton:

I'm pleased to inform you that your manuscript has been deemed suitable for publication in PLOS ONE. Congratulations! Your manuscript is now with our production department. 

Kind regards, 

on behalf of

Dr. Christopher M. Danforth 

Academic Editor

PLOS ONE